# Sodium and Human Health: What Can Be Done to Improve Sodium Balance beyond Food Processing?

**DOI:** 10.3390/nu16081199

**Published:** 2024-04-18

**Authors:** Angelo Tremblay, Marie-Pascale Gagné, Louis Pérusse, Catherine Fortier, Véronique Provencher, Ronan Corcuff, Sonia Pomerleau, Nicoletta Foti, Vicky Drapeau

**Affiliations:** 1Department of Kinesiology, Faculty of Medicine, Université Laval, Québec, QC G1V 0A6, Canada; louis.perusse@kin.ulaval.ca (L.P.); catherine.fortier@crchudequebec.ulaval.ca (C.F.); vicky.drapeau@kin.ulaval.ca (V.D.); 2Institute of Nutrition and Functional Foods, Université Laval, Québec, QC G1V 0A6, Canada; marie-pascale.gagne@fsaa.ulaval.ca (M.-P.G.); ronan.corcuff@fsaa.ulaval.ca (R.C.); sonia.pomerleau@fsaa.ulaval.ca (S.P.); nicoletta.foti.1@ulaval.ca (N.F.); 3Centre Nutrition, Santé et Société (NUTRISS), Institute of Nutrition and Functional Foods, Université Laval, Québec, QC G1V 0A6, Canada; veronique.provencher@fsaa.ulaval.ca; 4Endocrinology and Nephrology Axis, CHU de Québec Research Center, Université Laval, Québec, QC G1V 0A6, Canada; 5School of Nutrition, Université Laval, Québec, QC G1V 0A6, Canada; 6Quebec Heart and Lung Institute Research Center, Quebec, QC G1V 4G5, Canada

**Keywords:** salt, blood pressure, cardiometabolic, obesity, appetite, fitness, eating behaviour

## Abstract

Sodium plays a key role in the regulation of water balance and is also important in food formulation due to its contribution to the taste and use in the preservation of many foods. Excessive intake of any essential nutrient is problematic and this seems to be particularly the case for sodium since a high intake makes it the nutrient most strongly associated with mortality. Sodium intake has been the object of recommendations by public health agencies such as the WHO and this has resulted in efforts by the food industry to reduce the sodium content of packaged foods, although there is still room for improvement. The recent literature also emphasizes the need for other strategies, e.g., regulations and education, to promote adequate sodium intake. In the present paper, we also describe the potential benefits of a global healthy lifestyle that considers healthy eating but also physical activity habits that improve body functionality and may help to attenuate the detrimental effects of high sodium intake on body composition and cardiometabolic health. In conclusion, a reduction in sodium intake, an improvement in body functioning, and educational interventions promoting healthy eating behaviours seem to be essential for the optimal regulation of sodium balance.

## 1. Introduction

Sodium is the main cation of the extracellular fluid and plays a major role in the regulation of the extracellular volume and water balance. It is also associated with thirst and drinking behaviours [1].

The importance of sodium is also noticeable in food science because of its numerous key technological roles that include the solubilization of proteins, the improvement of water retention, and the inhibition of microbial growth [2,3,4,5]. With respect to food preparation and culinary art, it influences salinity, increases food flavours, and improves food textures. Finally, even today, its role in food preservation remains essential in the formulation of some foods. Globally, these properties confer to sodium the profile of a nutrient that is at risk of being overconsumed.

High dietary sodium intake has been documented in many countries and the recommendation to reduce its consumption is a matter of consensus among public agencies. In the Global Burden of Disease Study 2010 [6], sodium intake was assessed in 66 countries representing 74.1% of the world population. The results showed that in 2010, the estimated sodium intake of 3.95 g/day was twice the reference intake of 2 g/day considered in this study, according to age, sex, and country. In addition, this study revealed that 1.65 million annual deaths from cardiovascular causes were attributable to sodium intake above the reference level. In a subsequent study by this research group [7], high sodium intake was found to be a leading dietary risk factor for deaths and disability-adjusted life-years.

Globally, these observations suggest that sodium is a nutrient that places people in conflict with themselves since its usefulness in food preparation and processing seems to be as important as the health-related risks resulting from its overconsumption. As described in this paper, the food industry has been solicited and has responded with various strategies to reduce the sodium content of processed foods. However, it is likely that complementary efforts by health professionals will be needed to achieve a level of sodium intake that is compatible with body homeostasis. This issue is discussed here by focusing on contributions which require the expertise of public health professionals, dietitians, kinesiologists as well as behaviour and microbiome specialists.

## 2. Impact of Governmental Interventions: The Canadian Experience

The reduction in the sodium content of processed foods is a matter of strong preoccupation for public health agencies in many countries [8,9,10,11,12,13]. The efforts currently deployed to reach this goal vary according to the source and the nature of the measures. These include national political commitment, voluntary measures, and compulsory measures [8]. In Canada, the governmental authority (Health Canada) implemented initiatives in 2007 via a national strategy that included the publication of sodium reduction targets for Canadians. In 2012, Health Canada proposed targets of voluntary decrease in collaboration with food producers and health professionals. This initiative was accompanied by campaigns for public education about sodium, research on this mineral, and monitoring of the sodium content in the Canadian food supply. Despite these targets being perceived as achievable, the assessment of the impact of these measures in 2017 only revealed a partial success since the observed reduction of 240 mg/day of sodium was still leaving the estimated daily intake (2760 mg/day) above the maximal targeted intake of 2300 mg/day [14]. This evaluation also showed that the reduction target was only achieved for 14% of food categories, whereas no significant change was observed for 48% of categories.

Since the voluntary approach used in Canada has not provided the expected outcome, more severe measures were envisioned from a regulatory standpoint. The main measure is the necessity to present a front-of-pack nutrition symbol on food packages to highlight the high content of sodium, in addition to two other targeted nutrients, i.e., saturated fat and sugar [15]. This regulation was imposed in 2022 and food manufacturers have up to 1 January 2026 to adjust their labelling where needed. The presentation of the symbol is determined by thresholds based on reference values or recommended amounts of nutrients and is also perceived as a potential interesting strategy of nutrition education for the general population. In this regard, data collected by the Food Quality Observatory of Université Laval show that processed meats, ready-to-eat soups and pizzas are the processed foods which are the most likely to require this symbol for their high content in sodium [16].

The Canadian experience is concordant with the systematic review reported by Santos et al. [17] about salt reduction around the world. These authors focused on changes observed in many countries following WHO recommendations in 2013 to decrease salt intake by 30% to reduce premature mortality from noncommunicable diseases by 25% by 2025 [17]. They reported that since 2014, there has been a significant effort to reduce salt intake. A substantial decrease (>2 g/day) was observed in three countries, a moderate decrease (between 1 and 2 g/day) in nine countries, and a slight decrease (<1 g/day) in five countries, whereas none had yet met the targeted decrease of 30% in salt intake. In a more recent WHO report, evidence was presented that sodium reduction efforts, including campaigns, progress in many countries of the world [8]. Nevertheless, progress has been slow, with only a few countries managing to decrease population sodium intake, yet none have reached the target. Consequently, there is contemplation about extending the target to 2030 [8].

In summary, the Canadian experience and that of other countries show that relevant population-based interventions to reduce sodium/salt intake resulted in partial success. In this regard, the Canadian experience suggests that voluntary measures are not sufficient to achieve a level of sodium intake that is compatible with public health recommendations. As described in the next section, the food industry is part of this endeavour although it is also facing constraints and limitations.

## 3. Potential Contribution of the Food Industry

The technological importance of sodium and its multidimensional functions in food processing explain its reduction or replacement complexity. Numerous scientific works have been conducted over the last years to limit the sodium content of processed foods. The strategies that are most frequently used include the reduction in added salt in formulations and the replacement of sodium chloride by alternative salts containing less or no sodium (e.g., KCl, MgCl_2_, CaCl_2_) or being structurally modified. Moreover, the addition of alternative ingredients such as flavour enhancers, e.g., herbs, spice seasonings and yeast extracts, and textural agents or the use of different processes are also part of the industrial strategies to favour a decrease in the sodium content of processed foods [11,18,19,20,21,22,23,24]. As described in Table 1, they can promote concrete solutions for a variety of foods.

Globally, these results show that it is possible to reduce the sodium content of processed foods (between 0 and 75%) and that the potential level of decrease varies mainly according to the food category. The reduction process should be implemented as a multifactorial concept because of the numerous functions of sodium in processed foods, i.e., taste, texture, food safety and preservation, depending on the food category. Another essential condition for the conceptual progress in this field is the rigour of available information that does not systematically provide adequate indications about the sodium reduction that is industrially achievable. As further discussed in this paper, the perception of consumers is also a major issue that has the potential to limit the progress of proposed actions. In this regard, it is relevant to emphasize the recent attempt to use flavour enhancers to increase salt perception in order to ultimately reduce the salt content of food [43]. These agents can be peptide structures or ingredients with salty and umami taste from natural sources. The umami taste is mainly attributed to glutamate content (glutamic acid) [44]. The main herbal seasoning as a salt alternative in food includes plants (i.e., salt grass, Chinese strawberry) as well as edible seaweed [45], allowing for a salt reduction of up to 43% [46]. In particular, the umami-flavouring properties of seaweeds are attributed to their high proportion of glutamic acid. Therefore, the glutamate content of seaweed may be the foundation for enhancing the umami taste in several food products [44]. Moreover, a commercial natural salt substitute developed from seaweed extracts (AlgySalt^®^) has been studied and reviewed as one of the promising salt-reducing methods in meat product manufacturing [47,48]. A similar taste-replacing strategy has been recently evaluated in the production of wheat bread [49]. The authors investigated the potential role of fermentation to replace the flavour of salt with those developed during traditional sourdough bread fermentation production. Those flavours have been attributed to glutamate and acid products (i.e., lactic, and acetic) accumulation induced by wild yeast and lactic acid bacteria fermentation [45,49,50,51]. Indeed, the results confirmed that the use of sourdough deepened the flavour of the bread to such an extent that a reduction of 0.9% in sourdough bread did not have an undesirable effect on the consumer [49]. These agents can be peptide structures or ingredients with an umami taste. Furthermore, the study of consumer-perceived saltiness has been complemented by experiments aiming at understanding how specific odours could increase the salt perception of less salted products [52,53].

The industrial interventions aiming at salt reduction are also guided by the intent to optimize the shape and distribution of salt in order to maximize its perception when chewing food. In this case, the main challenge is to decrease the size of salt particles which induces an increase in salt perception by taste receptors [54]. This effect results from a faster increase in oral salt concentration provided by the greater surface of salt exposure of small particles [55]. However, despite the promising potential of this process, the industrial complexity of producing small salt particles in a homogenous manner currently remains economically non-viable [33].

In summary, the available literature shows that there are numerous theoretical solutions to reduce the salt content of processed foods. For the food manufacturer, the challenges to consider include the guarantee of a healthy product with a stable quality over time, production at a viable cost, and the achievement of satisfactory taste properties for consumers. In this context, many food processors do not have yet a complete solution, but more efforts are being deployed to reach this objective.

## 4. What Can Be Done beyond Food Processing to Reduce the Sodium Content of Food?

Two categories of solutions are generally considered to deal with a problem such as the excess sodium intake. As described above, the first one consists of reducing the intensity of an external stimulus, i.e., policy and food environments, which is well exemplified by the significant efforts deployed up to now by public health professionals and some food producers to decrease the sodium content of processed foods [56]. The second category of solutions shares the preoccupations of lifestyle medicine and aims to increase individual capacities to deal with an external stimulus. They may still involve food processors but they mostly rely on the expertise of dietitians and other health professionals.

## 5. Modulation of Salt Taste of Food According to Consumer Preferences

Processed foods largely contribute to the sodium intake of people [12,57,58,59]. Specifically, prepared and prepackaged foods provide about 75% of dietary sodium intake [12,60]. Among them, bread, processed meats, and cheese are the greatest sources of sodium, not only because of their high sodium content but also because they are largely consumed by the population. In this context where it is difficult to make substantial adjustments to the composition of these foods, some investigations have targeted consumer habituation as an approach to reduce sodium intake. In a recent Australian study, it was found that a reduction of 25% of the salt in bread, being among the best sellers in this food category, did not affect sales [61]. Concordant results were observed in another study showing that a decrease of 40% of the sodium content of breads did not seem to alter their acceptability in Brazilian consumers [62]. This is also in agreement with a study conducted in Tunisia that confirms the feasibility of reducing the salt content of bread without an unfavourable appreciation from consumers [63]. Globally, these studies suggest that even if sodium is essential for organoleptic, microbiological, and technological aspects of bread preparation [64], it is possible to develop interesting products by carefully considering consumer preferences.

## 6. Adopting the DASH Diet

The Dietary Approaches to Stop Hypertension (DASH) diet was proposed by the National Heart, Lung, and Blood Institute in 1977 to control the blood pressure (BP) of hypertensive patients. The DASH eating plan encourages the consumption of potassium-rich vegetables and fruits, whole grains, poultry, fish, and nuts, and reduces sodium and saturated fat intake [65]. There is strong evidence that the DASH dietary pattern can lower blood pressure (BP) [66,67], including in patients with type 2 diabetes and those with hypertension [68,69]. Numerous clinical studies, such as the DASH sodium study [66], the PREMIER trial [70], and the OmniHeart trial [71], as well as other studies, demonstrate the beneficial effects of the DASH diet to lower sodium intake and manage hypertension.

A recent systematic review and meta-analysis confirmed the efficacy of the DASH diet. This study identified 30 RCTs with 5545 participants that investigated the BP effects of the DASH diet compared with a control diet in adults with or without hypertension. The adoption of the DASH diet was accompanied by significant BP reduction in all adults independently of their hypertension conditions. However, higher daily sodium intake and younger age enhanced the BP-lowering effect of the intervention [72]. The evidence is thus very strong on the efficacy of this diet in lowering sodium consumption and decreasing BP. However, because the DASH diet requires daily consumption of 2300 mg of sodium (equivalent to 5.8 g of sodium chloride) or less [66], adherence to this diet may be difficult, particularly in some populations with a high preference for sodium—e.g., Chinese populations. Also, it is important to implement the DASH diet through nutritional counseling and other strategies that could help individuals decrease their sodium intake such as technological use and government policy support.

## 7. Education on Salt Alternatives

Educating individuals on how to use herbs, spices, and salt-free seasonings can enhance food flavour without adding sodium. This strategy can help satisfy the palate without the need for high-sodium options. A recent study modelled the influence of using herbs/spices as flavour enhancers when reducing overconsumed dietary components such as sodium (but also saturated fats and added sugar) in commonly consumed foods and evaluated the acceptance of these flavour-enhanced reformulations [73]. Results suggest that using herbs/spices to create flavour-enhanced recipes can potentially reduce sodium intake and is acceptable to consumers.

In clinical practice, replacing salt with low-sodium-salt substitutes (LSSS) can also potentially simultaneously decrease sodium intake and increase potassium intake. Benefits of LSSS include their potential BP-lowering effect and relatively low cost. However, there are concerns about possible adverse effects of LSSS, such as hyperkalemia, particularly in people at risk, for example, those with chronic kidney disease (CKD) or taking medications that impair potassium excretion. A recent systematic review documented the effect of replacing salt with LSSS in RCT studies, including 34,961 adults and 92 children. The smallest RCT included 10 participants and the largest had 20,995 participants. The proportion of sodium chloride replacement in the LSSS interventions varied from approximately 3% to 77%. The review concluded that compared to regular salt, LSSS reduces BP, non-fatal cardiovascular events and cardiovascular mortality slightly in adults and, more specifically, in those with elevated BP. However, there is a lack of evidence in children, pregnant women, and people in whom an increased potassium intake is known to be potentially harmful, limiting conclusions on the safety of LSSS in the general population [74].

## 8. Using Digital Strategies to Reduce Salt Intake

Digital strategies can be instrumental in helping individuals select low-sodium foods and adopt a low-sodium diet. For example, a study assigned 121 participants to three groups: a control group, a WhatsApp group, and an Electronic Brochure group, for 6 weeks. The WhatsApp group received weekly messages introducing new topics on reducing sodium intake, with reminders every 3 days. The Electronic Brochure group received an electronic brochure via email every 2 weeks, each covering a new topic. This study showed that the United Arab Emirates (UAE) population responded well to a salt reduction initiative using digital platforms over 6 weeks. The WhatsApp group saw a greater decrease in salt intake compared to the Electronic Brochure and control groups. Moreover, this reduction in salt intake led to a significant decrease in the proportion of participants exceeding the WHO’s recommended sodium intake by 10% in the WhatsApp group [75].

Smartphone apps also offer support for adherence to the DASH diet. A review of seven apps found that, despite most having significant quality and data security issues, NOOM and DASH To TEN had adequate quality and security, making them suitable for DASH diet self-management [76]. Yet, the clinical effectiveness of these apps remains unproven. Five studies (three RCTs and two pre-post studies) including 334 participants examined DASH mobile apps. Although all studies indicated a positive trend in using DASH smartphone apps, the three RCTs were highly biased. Although all RCTs assessed the outcome (BP level and DASH diet adherence) with comparable methods across intervention groups, they lacked outcome evaluation blinding, leading to a high risk of bias in the “outcome measurements” domain. The conclusion was that there is weak emerging evidence for a positive effect of DASH smartphone apps in supporting self-management, improving DASH diet adherence, and consequently lowering blood pressure [77]. The Nourish study, a randomized controlled trial testing a commercially available smartphone application to improve adherence to the DASH diet among adults with hypertension, might prove the potential impact of apps in adhering to the DASH diet and decreasing dietary sodium [78]. Additionally, a study by Sookrah et al. [79] underscores the potential of AI technology in applying the DASH diet with a system that generates dietary recommendations based on user input and DASH nutritional guidelines.

In conclusion, while digital strategies and apps show promise in promoting low-sodium diets and supporting DASH diet adherence, ongoing research and improvements in app quality and security are essential for maximizing their effectiveness and reliability in public health initiatives.

## 9. Modification of Certain Eating Behaviours

***Mindful eating practices:*** Despite the limited literature on this topic, some evidence suggests that promoting mindful eating can help individuals become more aware of their eating habits and food choices. A randomized clinical trial involving 201 participants revealed that, at a 6-month follow-up, participants in the Mindfulness-Based Blood Pressure Reduction (MB-BP) program had a statistically significant improvement in their interoceptive awareness score by 0.54 points and their DASH score by 0.62 points compared to the control group [80]. These findings indicate that the 8-week MB-BP program can enhance adherence to DASH eating patterns, suggesting that mindful eating may lead individuals to pay more attention to their food selections and eating habits, thereby encouraging the choice of lower-sodium options.

***Improving emotional regulation****:* Emotional eating can trigger disinhibited eating behaviours, leading individuals to consume high-sodium comfort foods. This type of eating has been significantly linked to a lower quality diet and increased consumption of fast food, commercially baked goods, pastries, sweets, and candy. Moreover, emotional eating has been associated with a higher intake of snacks and fast food, resulting in increased sodium consumption [81,82]. It can be hypothesized that adopting emotional regulation strategies may enable individuals to find healthier ways to manage their emotions, decreasing their dependence on high-sodium foods for emotional support.

Thus, integrating mindful eating practices and emotional regulation strategies into dietary interventions could have a beneficial impact on sodium intake reduction. By fostering a greater awareness of eating habits and emotional triggers, individuals may be more likely to make healthier food choices, including those with lower sodium content.

The most effective solutions for reducing sodium intake will likely be multifaceted, combining individual, educational and behavioural interventions within a clinical context with population-based strategies that focus on supporting food environments. Such strategies include salt reduction policies aimed at reformulating foods by the industry, enhancing food labelling, implementing taxes or subsidies, and launching communication campaigns. These approaches have proven highly effective in reducing sodium intake across the population [83].

## 10. Effects of Physical Activity and Aerobic Fitness

Population studies have clearly established that high sodium intake is positively associated with obesity [84,85], especially in females [86,87]. We recently confirmed this association and reported for the first time an interaction effect showing that the relationship between sodium intake and body composition varies according to levels of aerobic fitness in females [88]. As illustrated in Figure 1, this interaction effect reflected a much less favourable profile of body composition and fat distribution in females classified in the high sodium–low fitness subgroup. To reinforce the demonstration of the validity of this effect, we performed mediation analysis, which is a tool used in research to address the question of the mechanisms or pathways by which an exposure causes an outcome [89]. Using this approach, we showed that both sodium intake and aerobic fitness are significant mediators of the association between genetic susceptibility to obesity assessed using a polygenic risk score of obesity and BMI or waist circumference [88]. This observation also raises the question as to why regular physical activity participation and/or aerobic fitness can exert a protective effect against some detrimental effects of high sodium intake. In an attempt to answer this question, we tested the mediating effect of aerobic fitness on the association between sodium intake (exposure) and body fatness (outcome). We observed that a low cardiorespiratory fitness mediated the association between sodium intake and percent body fat with a mediating effect reaching 24% (*p* = 0.007). These results suggest that the detrimental effects of sodium intake on body fatness are partly explained by aerobic fitness.

A partial answer to this question comes from studies in sports sciences showing that prolonged vigorous exercise can induce a substantial sweat loss of salt. Thus, in a subgroup of marathon runners tested by Lara et al. [90], mean sweat salt losses were equivalent to 3.5 g NaCl per litre of sweat. Experimental data also demonstrated that sweat sodium losses are increased under high dieting salt conditions [91] and suggest that sweat may play a role in the regulation of sodium balance in humans.

Another explanation of the benefit of an active lifestyle on sodium balance is related to a potential impact on salt sensitivity. This was investigated by Rebholz et al. [92], who tested the effect of physical activity (PA) habits on the responses of blood pressure from a low-sodium to a high-sodium intervention. As expected, the switch from a low- to a high-sodium diet increased blood pressure in all PA subgroups. However, this response was attenuated in persons of the higher quartile of physical activity who displayed an adjusted odds ratio of high salt sensitivity of 0.66 compared to participants classified in the lowest quartile. As discussed by the authors, the protective effects of increased levels of physical activity might be explained by several potential mechanisms such as reduction in insulin resistance, improvement of endothelial function and inhibition of sympathetic nervous system activity.

Kitada et al. [93] recently described a mechanism aiming at water conservation in the context of high salt intake that may be linked with glucocorticoid-driven muscle catabolism. In this regard, the previous demonstration that urea reincorporation into protein is increased during and after exercise [94] and that exercise protects against muscle atrophy and glucocorticoid-induced muscle wasting [95] can also represent relevant beneficial effects of regular PA participation.

In summary, the evidence described in this section suggests that beyond industrial, behavioural, and nutritional actions favouring a reduction in salt intake, improving levels of physical activity and aerobic fitness is also part of what should be carried out to promote an adequate sodium balance. It seems to promote some metabolic changes facilitating the regulation of sodium balance, especially in the context of high sodium intake. The specific mechanisms underlying these benefits need to be clearly established but current evidence justifies further relevant investigation.

## 11. Intervening on the Gut Microbiome

Recent research has associated variations in the gut microbiota and the development of cardiometabolic dysfunctionalities and diseases. This conforms with our experience which showed that the supplementation of a Lactobacillus Rhamnosus probiotic in the context of a diet-based weight loss program accentuated the benefits on body composition, cardiometabolic markers as well as eating and mood-related behaviours [96,97]. Recently, Wilck et al. [98] documented the role of the depletion of Lactobacillus species of the gut microbiome in the interaction effect between high sodium intake and T cell phenotype on blood pressure. In another recent study, Wyatt and Crowly [99] tested the impact of a Lactobacillus treatment on the development of salt-sensitive hypertension in mice. As expected, a high-salt diet induced an increase in blood pressure that was attenuated in animals also receiving the probiotic supplement. Thus, even if further research is needed to study this issue, these observations provide optimism about interventions targeting the microbiota to improve sodium-related homeostasis.

## 12. Conclusions

Sodium plays a vital role in body homeostasis but, as for other nutrients, its overconsumption is associated with severe cardiometabolic problems. In this case, the management of optimal nutrient intake is more complex and difficult because of the importance of sodium in food preparation, processing, and preservation. Following recommendations of governmental and public health agencies, part of the food industry has responded and some progress has been achieved to decrease the sodium content of processed foods and it is anticipated that additional progress might be achieved due to technological developments. However, the scientific literature described in this paper suggests that an optimal sodium balance cannot be reached without concomitant interventions favouring healthy lifestyle habits. These include nutritional counselling and behaviour modifications promoting healthy eating and a consequent indirect reduction in sodium intake. Changes in non-nutritional factors such as vigorous physical activity participation are also relevant since they may accentuate sodium losses and attenuate the negative effect of high sodium intake. Along these lines, recent research also emphasized the potential benefits resulting from relevant modifications of the gut microbiota. In brief, the optimal success will likely come through a reduction in sodium exposure, an improvement of body functionality and educational interventions to facilitate the regulation of sodium balance.

## Figures and Tables

**Figure 1 nutrients-16-01199-f001:**
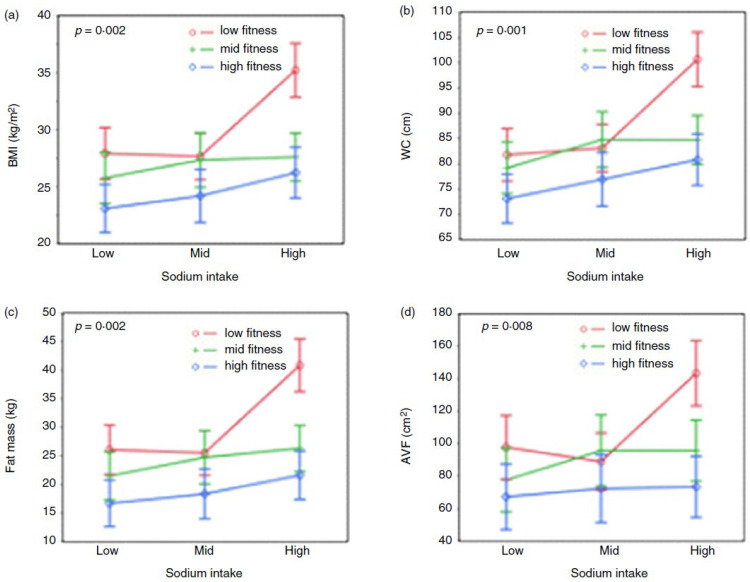
Effect of sodium intake on body composition in females according to fitness level. The figure presents the age- and reporting status-adjusted values for BMI (panel **a**), waist circumference (panel **b**), fat mass (panel **c**) and abdominal visceral fat (panel **d**). Groups of sodium intake and fitness are defined based on sex-specific tertiles of the age-adjusted data of each variable. The *p* value reported is the one corresponding to the effect of interaction. Reprinted from Tremblay et al., 2023 [88].

**Table 1 nutrients-16-01199-t001:** Summary of different strategies to reduce the sodium content of various food and their reduction targets.

Strategy		Examples	Reduction Rate Achieved (Range)	Food Categories Where the Strategy Is Applied	References
Reduction in sodium content in the formulation	Reduction in salt content added to formulations	Radically reduce added salt. Gradually reduce added salt	Reduction between 0 and 50% in sodium content	Processed meat and poultry products, bread, crackers, snack foods, salad dressings, soups, sauces, broths	[19]
			Up to 40%	Ready-to-eat meals	[25]
	Substituting high-sodium ingredients with low-sodium alternatives	Substitute versions of crushed tomatoes, sauces, broths, regular spices with their reduced sodium alternatives			
Substitution of salt with alternatives containing little or less sodium	Use of potassium chloride (KCl), calcium chloride (CaCl_2_), magnesium chloride (MgCl_2_)		0–50% (KCl)	Bread	[19,26]
			30–75% (KCl and mixtures)	Dairy products (cheese)	[27]
			20–33% (KCl and mixtures)	Processed meat products (sausages)	[27]
			30–50% (KCl and mixtures)	Seafood (smoked fish, sauce)	[27]
			Up to 40% (KCl)	Soups	[28]
			18% (KCl + MSG)	Soups	[29]
			Up to 75% (KCl + soy sauce odor)	Soups	[30]
	Use of mineral salt mixtures, sea salt				
	Use of structurally modified salts	Use micronized, flaked, liquefied, encapsulated salts	Up to 65%	Potato chips	[31]
			n.d. NaCl trapped in duckweed	Soup noodles	[32]
			Up to 25–50%	Crackers	[20]
			Up to 33%	Meat products	[33]
Addition of ingredients to formulations to compensate for the effects associated with sodium reduction	Addition of flavour enhancers/maskers	Add yeast extract, amino acids, algae, vegetable extracts or powder (mushrooms, tomatoes, etc.), spices to formulations	Up to 65%	Potato chips	[31]
			12.25%	Potato chips	[34]
			18–57%	Bread	[19]
			22%	Processed meat products (ham)	[27]
			Up to 40%	Soups	[28]
			32.5% (MSG)	Soups	[35]
			20% (yeast extract)	Soups	[36]
			18% (KCl + MSG)	Soups	[29]
			n.d. (fermented soy sauce)	Dressings	[37]
	Addition of textural agents	Add dietary fibre, proteins, food gums, enzymes to formulations	Up to 20%	Bread	[26]
				Processed meat products	[38]
	Addition of antimicrobial agents	Add to formulations weak organic acids and its salts, nitrates/nitrites, phosphates, modified vinegar and fermentation products, biopreservative microbial cultures and bacteriocins, essential oils, plant extracts, phenolic compounds	10–50% using bioprotective cultures	Various products (miso/soy sauce, cheeses, sausages)	[27]
Modification or addition to manufacturing processes to compensate for the effects associated with sodium reduction	Use of conservation processes	Use pasteurization, cold pasteurization (high hydrostatic pressure (HPP)), refrigeration/freezing, under vacuum and other emerging technologies (microwave, electric fields/pulsed light, ultrasound)	n.d. Freezing/deep-freezing techniques	Frozen meals	[39,40]
	Use of active packaging for conservation	Use modified atmosphere, vacuum, antimicrobial packaging with intelligent indicators	n.d. (active packaging)	Processed meat products	[41,42]

n.d.: not determined.

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
