# Peer review of "Sodium and Human Health: What Can Be Done to Improve Sodium Balance beyond Food Processing?"

_nutrients, 2024, doi:10.3390/nu16081199_

Round 1
Reviewer 1 Report
Comments and Suggestions for Authors
The paper entitled "Sodium and human health: what can be done to improve sodium balance outside of food processing?" deals with the major public health problem of increased salt intake in the population.
I suggested the authors to research the national campaigns of individual countries on salt intake and list as a possible solution. For example, Croatia has introduced national legislation that prescribes the highest permitted level of salt concentration in bakery products, in order to limit producers from excessive addition of salt to products.
The only effective way to reduce salt intake among the population and to reduce health risks is to influence through national campaigns and legislation in food production. I highly recommend authors to consider my comments.
References are up to date and topic is very important to reduce health risks among the population.
Author Response
We extend the paragraph preceeding the summary of Section 2 (line 97) by adding the following sentence: “In a more recent WHO report, evidence was presented that sodium reduction efforts, including campaigns, progress in many countries of the world (ref 18).”
Ref 18: WHO global report on sodium intake reduction. Geneva, World Health Organization 2023.
Reviewer 2 Report
Comments and Suggestions for Authors
This is an interesting and valuable article regarding the various strategies employed for reducing sodium intake.
Too few was said in the Introduction about why the desire for sodium is so strong in humans as reflected by the multiple mechanisms for recuperating and retaining sodium.
One strategy to reduce sodium intake is to use spices (such as coriander, cumin, thyme, basil etc. as well as onion, garlic etc.) for food flavoring - the authors have not mentioned it (or have they?). Are there no studies regarding this strategy?
Author Response
We are aware that Paragraph 2 of the Introduction presents a very short overview about why the desire for sodium is so strong. This was done on purpose since relevant mechanisms are further considered and described in subsequent sections of the paper. Considering this point, we would be inclined not to modify the introduction in order not to be in duplication with these sections.
As suggested by the reviewer, we now refer to herbs and spices in the text even if this information is already presented in Table 1. Specifically, we propose the following insertion in line 111 following the word “enhancers”: “…, e.g. herbs, spice seasonings and yeast extracts, …”.
Round 2
Reviewer 2 Report
Comments and Suggestions for Authors
One of my concerns was that "Too few was said in the Introduction about why the desire for sodium is so strong in humans as reflected by the multiple mechanisms for recuperating and retaining sodium."
The authors' answer was that "This was done on purpose since relevant mechanisms are further considered and described in subsequent sections of the paper." I beg the authors to precisely indicate the paragraphs were these mechanisms are discussed - which are the lines where I can find these paragraphs?
My other question also remained unanswered: "Are there no studies regarding this strategy?" (I mean the strategy of using spices to reduce salt cravings).
Author Response
The following information is presented about the two points raised by the reviewer:
- Point 1: Mechanisms underlying the strength of the desire for salt
First, it is relevant to remind that the main focus of our paper is what can be done to improve sodium balance beyond food processing. In this context, which is not a priori the one of a physiological paper, it is not relevant and/or possible to put a strong emphasis on mechanisms promoting the desire for salt. However, as indicated to the reviewer in the first revision of the paper, we have tried to refer to mechanisms in each section where it was relevant to make such a reference. That being said, we present here the specific references to lines or sections where we have tried to consider mechanisms (viewed in a broad sense) , including the introduction:
- Lines 33-35: We refer to some regulatory properties of sodium and its associations with drinking behaviors.
- Lines 151-154: In this case, we present information about the perception of taste receptors which varies according to the size of salt particles.
- Section 5: This section provides evidence that taste perception is a trainable phenomenon.
- Lines 205-208: Evidence is presented here to suggest that the strength of sodium preference may limit the adherence to a regimen such as the Dash diet.
- Section 8 documents the fact that sodium desire seems to be behaviorally modifiable via the use of digital strategies.
- Sections 10 and 11: A uniqueness of this paper is to document mechanisms by which sodium balance can be modified without specifically targeting the desire for sodium. Here, the focus is put on mechanisms related to aerobic fitness and the gut microbiome.
- Point 2: Spices to reduce salt cravings
The reviewer raised an important question that is not answered by the available literature. This perception has been reinforced by a specific search of literature that was done following the receipt of the reviewer's comment. We used key words such as "salt craving", "salt craving reduction", "salt craving reduction spices", and "spices salt craving reduction". In fact, the paper that is the most frequently detected by this search is the one of Morris et al. (2008) which presents an interesting description of salt craving, but which does not consider the impact of spices to reduce salt cravings in the context of free-living conditions.